# Wooden-Tip Electrospray Mass Spectrometry Characterization of Human Hemoglobin in Whole Blood Sample for Thalassemia Screening: A Pilot Study

**DOI:** 10.3390/molecules27123952

**Published:** 2022-06-20

**Authors:** Tingting Huang, Ting Huang, Yongyi Zou, Kang Xie, Yinqin Shen, Wen Zhang, Shuhui Huang, Yanqiu Liu, Bicheng Yang

**Affiliations:** 1Maternal and Child Health Affiliated Hospital of Nanchang University, Nanchang 330006, China; htt19871212@126.com (T.H.); huangting626@126.com (T.H.); zouyongyi@gmail.com (Y.Z.); ragexp@126.com (K.X.); syq2657454560@126.com (Y.S.); zw595581428@126.com (W.Z.); 2Jiangxi Key Laboratory of Birth Defect Prevention and Control, Jiangxi Maternal and Child Health Hospital, Nanchang 330006, China

**Keywords:** thalassemia, human hemoglobin, wooden-tip electrospray ionization, multiply charged ions, mass spectrometry, multiply charged ion

## Abstract

Traditional analytical methods for thalassemia screening are needed to process complicated and time-consuming sample pretreatment. In recent decades, ambient mass spectrometry (MS) approaches have been proven to be an effective analytical strategy for direct sample analysis. In this work, we applied ambient MS with wooden-tip electrospray ionization (WT-ESI) for the direct analysis of raw human blood samples that were pre-identified by gene detection. A total of 319 whole blood samples were investigated in this work, including 100 α-thalassemia carriers, 67 β-thalassemia carriers, and 152 control healthy samples. Only one microliter of raw blood sample was directly loaded onto the surface of the wooden tip, and then five microliters of organic solvent and a high voltage of +3.0 kV were applied onto the wooden tip to generate spray ionization. Multiply charged ions of human hemoglobin (Hb) were directly observed by WT-ESI-MS from raw blood samples. The signal ratios of Hb chains were used to characterize two main types of thalassemia (α and β types) and healthy control blood samples. Our results suggested that the ratios of charged ions to Hb chains being at +13 would be an indicator for β-thalassemia screening.

## 1. Introduction

Human thalassemia is an inherited blood disorder that causes patients to have less hemoglobin (Hb) than normal. The disease is commonly found worldwide, affecting part of the population in southern China and other Southeast Asian countries [1,2]. There are two main types of thalassemia, alpha thalassemia (α-thalassemia) and beta-thalassemia (β-thalassemia), in patients with thalassemia who are characterized by reduced α or β globin chain synthesis [3,4]. The absent production of α or β globin chains results in reducing the composition of Hb and the production of red blood cells and generating anemia symptoms, all of which are life-threatening conditions. Blood transfusions can usually improve patients’ conditions [5]; however, there is currently no effective clinical method for thalassemia treatment, except stem cell transplantation and gene therapy [6]. Therefore, early detection and prevention play a key role in reducing thalassemia incidence through thalassemia screening.

Recently, there are many diagnosis methods for thalassemia screening [7,8], for example, routine blood analysis by mean corpuscular hemoglobin (MCH), mean corpuscular volume (MCV), erythrocyte osmotic fragility test (EOFT), serum iron test (SIT), Hb electrophoresis measurement (HEM), isoelectric focusing (IEF), liquid chromatography (LC), and next-generation sequencing (NGS). Among these methods, MCH and MCV are commonly used for thalassemia screening; however, these two methods are not specific enough. EOFT and SIT are typically used to identify microcytic hypochromic anemia with decreased MCH and MCV. HEM has been used for thalassemia screening, as this method can measure different electrophoretic behaviors of Hb tetramers. However, the principle of HEM is based on the structural integrity of intact Hb tetramers, and thus HEM could be affected by hemolysis and degradation of whole blood samples during storage. The combination of IEF and HPLC techniques could facilitate the identification of most known hemoglobinopathies. However, there is a limitation as it requires time-consuming and professional data interpretation, which is not amenable to thalassemia screening in wide clinical tests [9].

Mass spectrometry (MS) can measure the mass/charge ratio (m/z) and ion intensity of various clinical samples. To date, various MS methods have been developed as reference techniques for clinical analysis [10,11,12]. Moreover, MS-based methods have also been applied for diagnosing hemoglobinopathies [13,14]. Due to the complicated matrixes in human blood samples, extensive sample preparation processes such as extraction and separation, which are labor intensive and time consuming, are usually required before LC-MS screening [7]. Therefore, reducing or removing the sample preparation and LC separation steps before MS analysis is highly demanded in clinical tests.

Ambient MS methods are of great interest for direct analysis of raw samples because analytes can be easily ionized with no or little sample pretreatment process [15,16,17,18]. Ambient MS is pioneered by desorption electrospray ionization (DESI), which allows direct determination of various analytes such as proteins from raw biological samples under ambient conditions [15]. Recently, various ambient MS methods have been developed for various applications [16]. Remarkable analytical properties were found in ambient ESI with different substrates [19]. Paper spray ionization is one of the powerful ambient ESI techniques using porous paper substrate to load and ionize raw samples. It is fact that paper spray is a great technological breakthrough in ambient ionization. Undoubtedly, paper spay can be used for the direct analysis of blood samples. Various excellent articles on blood analysis using paper spray have been published [20,21,22]. Similar to paper spray, electrospray ionization with a wooden tip (WT-ESI) is an ambient ESI method with the advantage that the ionization process occurs under ambient conditions in which the raw samples can be easily accessible during analytical processes [23]. WT-ESI has been successfully developed to analyze proteins, clinical samples, and other complex biological samples as a useful tool for direct analysis with high sensitivity and high specificity [24,25,26,27,28]. In the WT-ESI method, disposable and low-cost wooden tips are quite convenient for direct sampling and analysis without any hardware modifications. The wooden tip can be directly mounted on commercial nanoESI devices by replacing the nanoESI emitter [19,23], which is greatly beneficial for non-expert users, especially in clinical use. In general, raw samples can be directly loaded on the surface of a wooden tip under the strong electric field without the use of any gas. The compounds of interest are extracted from raw samples and then electrospray ions were directly generated from the wooden tip-end for MS analysis. In particular, WT-ESI-MS can also be used for the quantification of target analytes from complex samples with only little sample preparation and no chromatographic separation, and the analytical performances, including the linear range, accuracy, precision, and sensitivity, were well acceptable for the direct analysis of real samples [24,29,30]. It is reported that different protein structures can be recognized by ambient MS [31]. In previous studies [32,33], WT-ESI-MS has been successfully used for protein analysis. Furthermore, the mobility of proteins in wooden tips could be characterized under ESI conditions [34]. Therefore, WE-ESI-MS is expected to characterize the protein structures from raw biological samples.

To the best of our knowledge, the ambient MS analysis of blood samples for the application in thalassemia detection has rarely been investigated. Herein, we would like to make a new attempt at the direct analysis of blood samples for thalassemia detection with ambient MS with WT-ESI. In this work, a total of 319 whole blood samples were examined, and we aim to explore the detection of α and β chains from whole blood samples by WT-ESI-MS. We assume that α and β chains in thalassemia and healthy blood samples would have different ion responses during the spray ionization process and MS detection. Hence, the signal ratios of α and β chains between thalassemia and healthy blood samples would be significantly different, and thus the ratios can be used for detecting α-thalassemia and β-thalassemia

## 2. Results

### 2.1. Direct Analysis of Whole Blood Samples

Figure 1 shows a schematic diagram (Figure 1a) and picture (Figure 1b) of an experimental setup for WT-ESI-MS analysis of human whole blood samples. Figure 2a shows mass spectrum of human Hb in organic solution (i.e., methanol/water/formic acid, 50/50/0.1, *v*/*v*/*v*) obtained by conventional ESI-MS. Multiply charged ions of α-chains and β-chains with denatured Hb were observed. A wide range of charged-state distributions (CSD) of α-chains from +10 to +21 was found under a Gaussian distribution. The charged stats at +14~+16 were dominant in the mass spectrum, while there was a relatively narrow CSD ranging from +11 to +18 for β-chains where the main peaks are +13, +14, and +15. These peaks were also found under Gaussian distribution. Particularly, free heme at m/z 616.18 was detected; the heme detection confirmed that Hb is totally denatured under the organic solvent. Low-background-noise (Figure 1c) solvent is generated a from blank wooden material that is similar to a previous work using wooden-tip ESI [32,33,34].

Figure 2b shows a typical mass spectrum of a healthy whole blood sample using WT-ESI-MS. Various peaks of blood lipids at m/z 700–900 were observed. Lipids are common biometrics in blood samples and play an important role in life science [33]. Moreover, the heme and multiply charged ions of the α-chain and β-chain were clearly observed ranging from m/z 900 to m/z 1600, indicating that WT-ESI-MS has high ionization efficiency for direct ionization of Hb from raw biological samples. Unlike the pure Hb standard with a wide CSD in solution, only the main peaks of Hb were observed from raw blood samples. It is noted that the CSD of the α-chain was ranging from +10 to +16, while the main CSD of the β-chain was found from +11 to +13. It is also found that these CSDs of α-/β-chains from raw whole blood samples are narrower than those in Hb standard.

In particular, it is important to note that the peaks of protein ions from raw blood samples are wider than those obtained from Hb standard, probably because Hb chains combined with other small ligands, salt, and molecules from biological matrices, which is a common phenomenon in direct ionization of proteins from untreated biological samples [33]. By using WT-ESI-MS, direct analysis of a single blood sample can be completed within 1 min. Overall, these results show that Hb can be directly detected from raw blood samples without sample pretreatment.

Figure 2c shows a typical mass spectrum of whole blood from an α-thalassemia sample using WT-ESI-MS. Similar to healthy blood samples, there are abundant peaks observed in the mass spectrum. Interestingly, the heme at m/z 616.17 was dominated in the mass spectrum at the base peak, while the peaks of lipids and multiply charged ions of α-chains (CSD: +10~+18) and β-chains (CSD: +11~+16) were clearly also observed. Figure 2d shows a typical mass spectrum of the whole blood from a β-thalassemia sample using WT-ESI-MS. Similar to other raw blood samples, although the relative abundance is different, there are abundant peaks of heme, lipids, and Hb chains (α-chain: CSD: +10~+16; β-chain: CSD: +11~+14) observed in the mass spectrum.

### 2.2. Signal Ratios of Main Protein Ions

Figure 3a shows signal ratios of α-chain to β-chain at the charged state of +11 obtained from α-thalassemia and healthy control samples. It was found that the ratios are at about 4.0 for both samples. However, there was no significant difference between thalassemia and healthy control samples. For the β-thalassemia samples at the charged state of +11 (Figure 3b), it was also found that there was no significant difference between thalassemia and healthy control samples.

When the α-chain and β-chain were at a charged state of +12, both α-thalassemia (Figure 3c) and β-thalassemia (Figure 3d) had no significant differences by comparing the healthy control samples. Interestingly, when the charged state was at +13, although there was no significant difference (Figure 3e) between α-thalassemia and healthy control samples, a significant difference was found between β-thalassemia and healthy control samples (*p* < 0.05), as shown in Figure 3f. Moreover, it was found that the ratios (median value: M; and average value: A) of healthy (A = 4.17, M = 4.06) and thalassemia (A = 4.69, M = 4.40) blood are different.

## 3. Discussion

The wooden tips were used for loading blood samples, and the organic solvent was loaded for extracting Hb chains to generate spray ionization. Multiply charged ions of Hb chains were obtained along with other biometrics. Considering multiply charged ions at different CSDs, each charged ion was compared in this work. As α chain and β chain are internal substances, the ratio would be stable for a normal blood sample, and thus the patients can be diagnosed. External and internal labels are not really needed. Our results suggested that the CDSs of α/β-chains at +13 (*p* < 0.05) and their ratios can be an indicator for β-thalassemia screening of a single blood sample. As 100 α-thalassemia, 67 β-thalassemia, and 152 healthy samples were analyzed in this work, some discrete data in Figure 3f are acceptable. In the electrophoresis process of chains of Hb, there are substantial significant differences obtained from β-thalassemia compared to α-thalassemia [35,36,37,38]. Therefore, by using a simple WT-ESI-MS method in this work, our results validated the results of the electrophoresis process of chains of Hb. Without any separation and sample pretreatment, discriminating β-thalassemia would be beneficial for developing a rapid diagnosis method based on ambient ionization methods in the future.

Hb is a tetramer with four polypeptide chains (α_2_β_2_) that included two alpha (α) and two beta (β) chains. The α-chain of Hb is comprised of 141 amino acids (molecular weight: 15,126 Da) [39], while the β-chain is comprised of 147 amino acids and has a molecular weight of 15,867 Da. Compared to native MS analysis of human Hb [40], the observation of chains and heme ambiguously shows that Hb tetramer was totally denatured in organic solution, and thus the basic sites such as basic amino acids and N-terminal residues were protonated in the ESI process [40]. As there are different sequences and masses between α-chains and β-chains, the two chains at the same charged states have different ionization responses in the ESI process. Therefore, it is a reasonable hypothesis that if there is a lack of an α/β-chain in an Hb sample, the signal ratio of α-chains to β-chains would be changed accordingly. It is also noted that there are different responses of lipids in thalassemia in this work, suggesting that the blood lipids would also be an indicator of the diagnosis and research of thalassemia [41]. However, more investigations are highly needed to further validate blood lipids for thalassemia screening.

Furthermore, the tetramer is denatured and Hb chains are unfolded in organic solution, and amino acid residues of Hb chains can be exposed to solution. More protons can be attached onto exposed amino acid residues when the unfolding degree of the proteins is greater. Increased charged states indicated that more amino acids were ionized, and changed Hb chains have stronger detectability than native Hb. Therefore, it is reasonable that there is a significant difference between thalassemia and healthy samples at a charged state of +13. Therefore, it can be expected that higher charged states would give higher significant differences. However, there is a limitation to generating higher charged ions of Hb chains from raw samples. Supercharging agents were reported to further elevate peptide and protein charge states in ESI [42], which would be beneficial for increasing the charged states of Hb chains. Moreover, compared to the Hb standard, there is relatively lower signal-to-noise for detecting raw blood samples. Surface-modified wooden-tip and microextraction methods might give some hints to improving the signal-to-noise and enhancing the detection of target analytes by reducing the matrix effects [26,27,43,44].

## 4. Materials and Methods

### 4.1. Chemicals and Materials

Human Hb was bought from Sigma. HPLC-grade Methanol was bought from the Chinese Chemical Reagent Co., Ltd. (Shanghai, China). Water used in this work was Milli-Q water. Wooden tips (toothpicks) were purchased from the local supermarket (Nanchang, China). The length of the wooden tip was cut to ~2.0 cm in this study. All wooden tips were washed with methanol and water and were dried (100 °C) before use. A total of 319 whole blood samples from 319 women (person, their ages were from 21 to 41 and the average age was 28) were collected into EDTA-K2 tube (Shanghai Zhengbang Medical Treatment Technology Co., Ltd., Shanghai, China), including 100 α-thalassemia carriers, 67 β-thalassemia carriers, and 152 healthy samples (without any gene type of thalassemia carriers). All blood samples were collected from Jiangxi Maternal and Child Health Hospital (Nanchang, China). α/β-thalassemia blood samples were firstly confirmed by polymerase chain reaction (PCR) and flow-through hybridization technology analysis (Hybribio Limited, Chaozhou, China) to assign the types of thalassemia carriers; the identification methods followed our previous studies [2,8].

### 4.2. WT-ESI-MS Analysis

Figure 1 shows the setup of wooden-tip ESI-MS. The wooden tips were cut to sharp points (tip size: 0.1 mm). The wooden tip was placed in the front of the mass spectrometer with distances of 0.8 cm horizontally from the wooden tip-end to the MS inlet via a nanoESI device (Thermo Fisher Scientific, Bremen, Germany). To load blood sample, whole blood samples (i.e., 1.0 μL) were directly loaded onto a wooden tip, and then organic solvent (5.0 μL of methanol/water/formic acid, 50/50/0.1, *v*/*v*/*v*), which is commonly used in spray ionization in ambient MS [45,46,47], was loaded onto the wooden tip in this work. With the application of a high voltage (3.0 kV) to wooden tip, spray ionization could be produced from wooden tip-end to acquire a mass spectrum. Mass spectra were acquired on an Orbitrap mass spectrometer (Thermo Fisher Scientific, Bremen, Germany). Data acquisition and instrumental control were conducted by using Xcalibur 3.0 software (Thermo Fisher Scientific, Bremen, Germany). The capillary temperature was set at 150 °C. The high voltage for spray ionization was supplied from the mass spectrometer to the wooden tip via a clip in a nanoESI device (Figure 1b). The acquisition speed was 4 scans/sec. Typically, data from the first 1 min were averaged to generate the mass spectra. All the mass spectra were directly obtained. To determine the signal ratios of each ion, the absolute intensity of each ion was obtained to compare in this work. Conventional ESI-MS was also performed to analyze Hb standard (10 μM) in organic solvent (i.e., methanol/water/formic acid, 50/50/0.1, *v*/*v*/*v*). ESI-MS conditions were referred to ESI-MS analysis of protein solution [48]. Briefly, the ionization voltage wat at 3.0 kV under positive mode; the capillary temperature was set at 150 °C; flowrate of protein solution was at 3.0 μL/min; sheath gas, aux gas, and sweep gas were set at 20 bar, 10 bar and 2 L/min, respectively. Each sample was analyzed three times to measure the *p*-value in this work. *p*-value calculation with a two-tailed test is used in this work.

## 5. Conclusions

In conclusion, we applied an ambient MS method with a WT-ESI approach to explore the differentiation of thalassemia using Hb chains. Without any sample pretreatment and separation, WT-ESI-MS can be used for the direct analysis of raw blood samples. A single sample can be completed within 1 min. In this work, a significant difference is found between β-thalassemia and healthy control samples. Our pilot study suggests that WT-ESI-MS and other ambient MS methods would be potential new clinical tools for thalassemia screening in the future. However, more investigations on differentiating different types of thalassemia are needed to further validate this method, and more mechanical studies of thalassemia screening by ambient MS are also needed to improve the applicability of thalassemia screening.

## Figures and Tables

**Figure 1 molecules-27-03952-f001:**
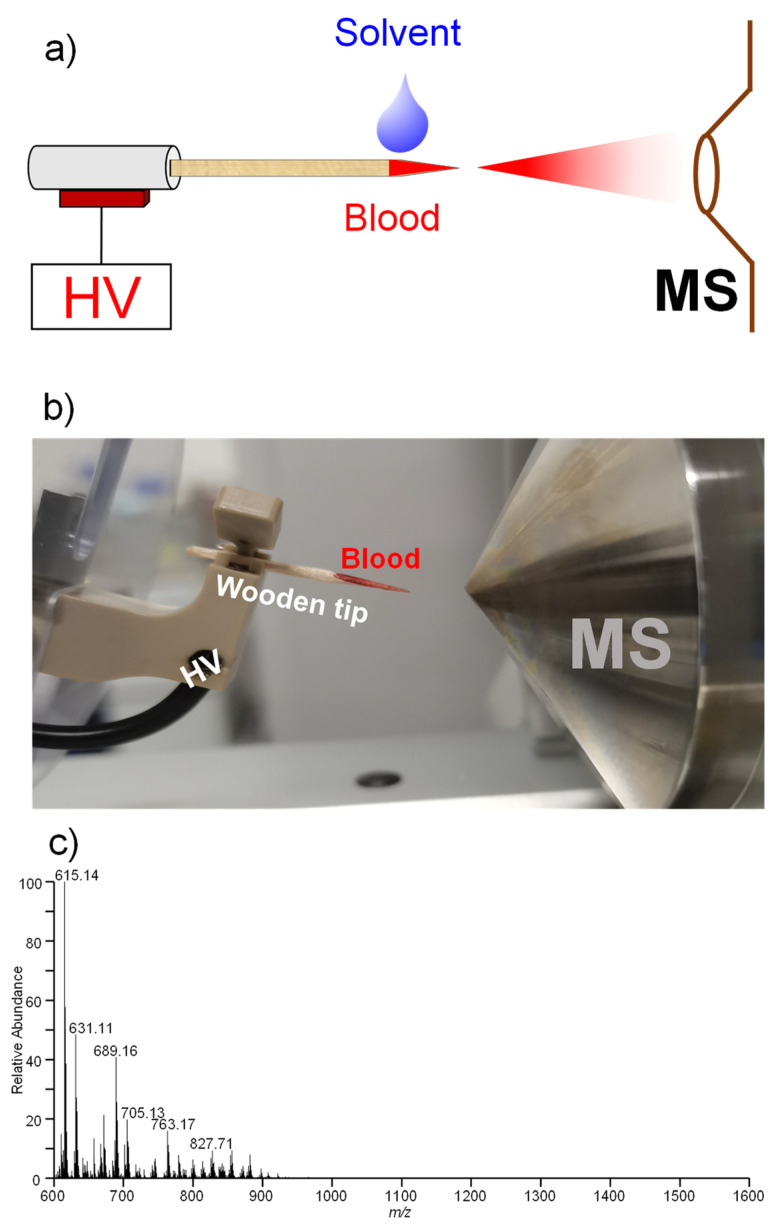
(**a**) Schematic diagram and (**b**) picture of WT-ESI-MS for direct analysis of blood samples; (**c**) background signal of a wooden tip with solvent only.

**Figure 2 molecules-27-03952-f002:**
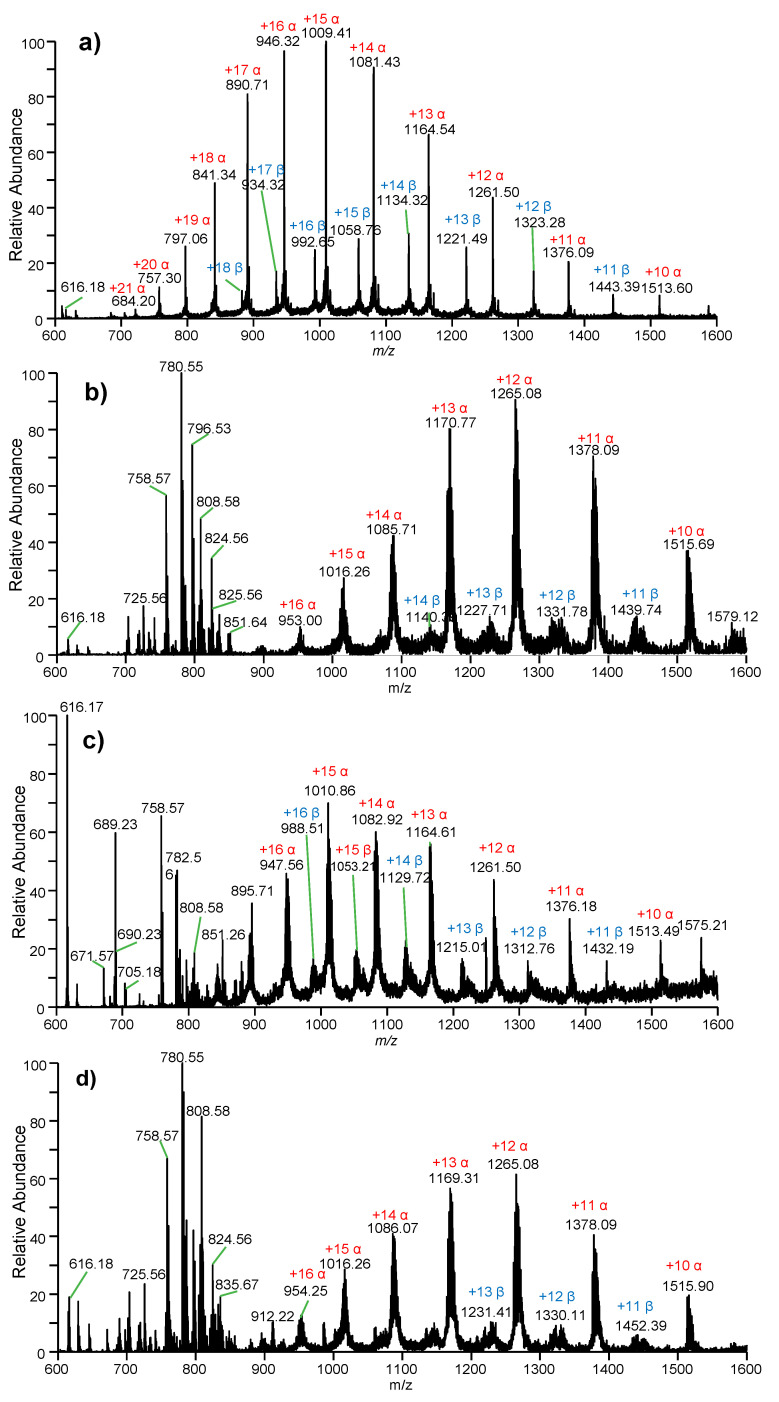
Mass spectra obtained by ESI-MS for the direct detection of human Hb from: (**a**) conventional ESI spectrum of pure Hb standard, (**b**) WT-ESI spectrum of healthy whole blood, (**c**) WT-ESI spectrum of α-thalassemia whole blood, and (**d**) WT-ESI spectrum of β-thalassemia whole blood.

**Figure 3 molecules-27-03952-f003:**
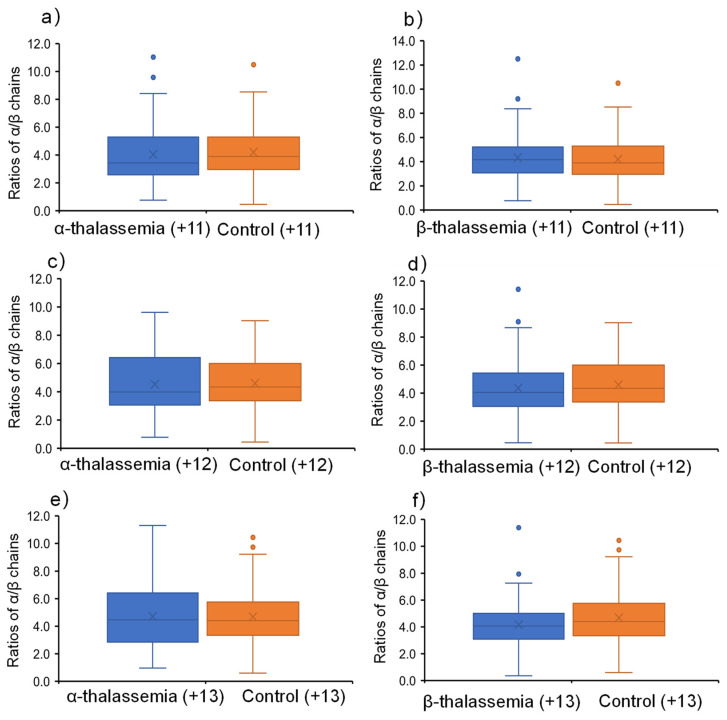
Ratios of Hb α-/β-chains from α-/β-thalassemia to healthly whole blood samples by wooden-tip ESI-MS under different charged states: (**a**) α-thalassemia (+11), (**b**) β-thalassemia (+11), (**c**) α-thalassemia (+12), (**d**) β-thalassemia (+12), (**e**) α-thalassemia (+13), and (**f**) β-thalassemia (+13).

## Data Availability

Not applicable.

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
