# Peer review of "Wooden-Tip Electrospray Mass Spectrometry Characterization of Human Hemoglobin in Whole Blood Sample for Thalassemia Screening: A Pilot Study"

_molecules, 2022, doi:10.3390/molecules27123952_

Round 1

Reviewer 1 Report

The authors conducted interesting experiments to analyze whole blood samples using the WT-ESI-MS method, but I feel the conclusions are premature. The last sentences of Abstract are not supported by the results. This method is not currently suitable for the diagnosis of the target disease and I think the hypothesis described in the introduction could not be verified.

Moreover, the subsequent application is questionable, as the statistical method could only show a small difference only in case beta-thalassemia. How could the method work for a single blood sample?

What do you think about the effect of hemolysis in case of your methods?

How were the ratios of Fig. 3. calculated? Which isotopic peak(s) was used for calculation? 

Spectrum of Fig 2a was measured by conventional ESI MS (line 107 page3) or by WT ESI MS (Figure legend)?

What could be the reason for the intense heme peak in the spectrum of Fig.2c ? Was this the case in all alpha-TH samples?

Overall, in its current form, I do not propose to accept the article.

Author Response

Piont 1: The authors conducted interesting experiments to analyze whole blood samples using the WT-ESI-MS method, but I feel the conclusions are premature. The last sentences of Abstract are not supported by the results. This method is not currently suitable for the diagnosis of the target disease and I think the hypothesis described in the introduction could not be verified.
Response: Thank you very much for your insightful comments and valuable suggestions. Perhaps there is a misleading presentation in the original manuscript. In the revised manuscript, we would like to emphasize that the diagnosis of the target disease has been verified by gene detection before using presented method in this work.

Piont 2: Moreover, the subsequent application is questionable, as the statistical method could only show a small difference only in case beta-thalassemia. How could the method work for a single blood sample?
Response: We agree with you that there is a small difference in case beta-thalassemia and healthy blood samples, because it is a fact that there are small differences of protein structures between healthy and thalassemia carriers, which have been well validated by electrophoresis. To clear reflect the differences between healthy and thalassemia carriers, we do not only use the p value, but also provide the ratios (median value: M; and average value: A) for differentiate healthy (A=4.17, M=4.06) and thalassemia (A=4.69, M=4.40) in the revised manuscript. Therefore, the ratios can be an indicators for measuring single blood sample.

Piont 3: What do you think about the effect of hemolysis in case of your methods?
Response: We are focusing on the protein analysis rather cell analysis, so the effect of hemolysis would not affect protein detection by this method. In the protein analysis, proteins were released from cells.

Piont 4: How were the ratios of Fig. 3. calculated? Which isotopic peak(s) was used for calculation?
Response: In addition, the peaks of protonated alpha or beta chains were used for calculation, the method and details for calculating the ratios of Figure 3 was also added in the revised manuscript.

Piont 5: Spectrum of Fig 2a was measured by conventional ESI MS (line 107 page3) or by WT ESI MS (Figure legend)?
Response: The spectrum of Figure 2a was measured by conventional ESI-MS, showing the Hb standard spectrum. We have corrected the figure legend.

Piont 6: What could be the reason for the intense heme peak in the spectrum of Fig.2c ? Was this the case in all alpha-TH samples?
Response: Thank you for your insightful comments. The intense heme peak in the spectrum of Fig.2c would be that the Hb was totally denatured. The heme response would also affected by the compositions of raw mixtrure such Hb chains in wooden-tip ESI-MS, because there are two alpha chains and two beta chains in intact Hb (α2β2) and other biomatirecls in blood samples. We have check all the data and found that there is fluctuant heme peak in alpha-TH samples, probably the complex matircies in blood sample, becase the resposne of small analyte are eaisly affected by matrixes; while ratios of protein chains would be more relaible, becuased the matice effect can be compsensated by the ratio.

Piont 7: Overall, in its current form, I do not propose to accept the article.
Response: In the revised manuscript, we have added more investigations, discussions and interpretation to improve the quality of this work, and hope you can find it can be accepted in Molecules.

Reviewer 2 Report

The paper titled “Wooden-Tip Electrospray Mass Spectrometry Characterization of Human Hemoglobin in Whole Blood Sample for Thalassemia Screening: A Pilot Study” describes the use of ESI-MS and the simple, direct, microliter volume of blood samples introduction technique with the use of regular wooden toothpicks. The described approach is very interesting from the clinical point of view and since this is the pilot study, it should be further evaluated  to prove its usefulness and reliability in the diagnosis of thalassemia. In my opinion this manuscript does not require major revision, but some minor remarks should be addressed. Also, editing should be improved, including English spelling, grammar and interpunction.

Some remarks:

What did the spectrum for blank – wooden tip with just solvent - looked like? Did the tip and the solvent generated noise signals that interfered with the analytical signal?

Figure 3f doesn’t look like there is a significant difference. Perhaps author could provide some numbers and more details of the statistical testing?

In what age range were the blood donors – the babies? How was the blood stored? Did the blood tubes contain any additional reagents like heparin, EDTA or others? Could the interfere with the Hb during the measurement in a significant way?

verse 33               “and thus led anaemia syndromes” – this part seems to have a grammar error double spaces, articles “a”,

verse 100            “difference” – should be “different”

verse 172            “and thus the patients can be found” – this sentence sounds as the patient was lost. Rephrase this, maybe with the word “diagnosed”.

verse 185            remove “long”

verse 200            Please rephrase this part “and the changeability of Hb chains have stronger detectability”

Author Response

Response to Reviewer 2 Comments

Point 1: The paper titled “Wooden-Tip Electrospray Mass Spectrometry Characterization of Human Hemoglobin in Whole Blood Sample for Thalassemia Screening: A Pilot Study” describes the use of ESI-MS and the simple, direct, microliter volume of blood samples introduction technique with the use of regular wooden toothpicks. The described approach is very interesting from the clinical point of view and since this is the pilot study, it should be further evaluated to prove its usefulness and reliability in the diagnosis of thalassemia. In my opinion this manuscript does not require major revision, but some minor remarks should be addressed. Also, editing should be improved, including English spelling, grammar and interpunction.
Response: Thank you very much for your positive comments and helpful suggestions. In the revised manuscript, we have addressed the minor remarks, and also improved the English thoroughly.

Point 2: Some remarks: What did the spectrum for blank – wooden tip with just solvent - looked like? Did the tip and the solvent generated noise signals that interfered with the analytical signal?
Response: We have added a new spectrum for blank wooden tip with just solvent, as shown in Figure 1c in the revised manuscript. The tip and solvent generate some background noise from the wooden materials. Similar to previous work using wooden-tip ESI, there are low noise signals at the high mass range for protein analysis. More literature is also added in the revised manuscript.

Point 3: Figure 3f doesn’t look like there is a significant difference. Perhaps author could provide some numbers and more details of the statistical testing?
Response: As suggested, more details of the statistical testing are provided in the revised manuscript. Briefly, 67 β-thalassemia and 152 healthy samples were investigated and P-value calculation for a two-tailed test is used.

Point 4: In what age range were the blood donors – the babies? How was the blood stored? Did the blood tubes contain any additional reagents like heparin, EDTA or others? Could the interfere with the Hb during the measurement in a significant way?
Response: We have added more details for the blood samples in the revised manuscript. The age range is from 22 to 34. After blood samples collected into the tube, EDTA was added. Our results, along with literature in protein-EDTA solution, show that EDTA dose not interfere the Hb detection using WT-ESI-MS.

Point 5: verse 33, “and thus led anaemia syndromes” – this part seems to have a grammar error double spaces, articles “a”, verse 100, “difference” – should be “different”, verse 172, “and thus the patients can be found” – this sentence sounds as the patient was lost. Rephrase this, maybe with the word “diagnosed”. verse 185, remove “long” verse 200, Please rephrase this part “and the changeability of Hb chains have stronger detectability”
Response: Corrected as suggested, thank you!

Round 2

Reviewer 1 Report

The reported method and approach are worth publishing, although I still consider the conclusion that it is suitable for distinguishing thalassemias to be premature.